# Diesel Particulate Extract Accelerates Premature Skin Aging in Human Fibroblasts via Ceramide-1-Phosphate-Mediated Signaling Pathway

**DOI:** 10.3390/ijms23052691

**Published:** 2022-02-28

**Authors:** Kyong-Oh Shin, Yoshikazu Uchida, Kyungho Park

**Affiliations:** 1Department of Food Science & Nutrition, Convergence Program of Material Science for Medicine & Pharmaceutics, Korean Institute of Nutrition, Hallym University, Chuncheon 24252, Korea; 0194768809@hanmail.net; 2LaSS Lipid Institute (LLI), LaSS Inc., Chuncheon 24252, Korea

**Keywords:** aging, ceramide, ceramide-1-phosphate, diesel particulate extract, matrix metalloprotease, skin

## Abstract

Both intrinsic (i.e., an individual’s body clock) and extrinsic factors (i.e., air pollutants and ultraviolet irradiation) accelerate premature aging. Epidemiological studies have shown a correlation between pollutant levels and aging skin symptoms. Diesel particle matter in particular leads to some diseases, including in the skin. Our recent study demonstrates that diesel particulate extract (DPE) increases apoptosis via increases in an anti-mitogenic/pro-apoptotic lipid mediator, ceramide in epidermal keratinocytes. Here, we investigated whether and how DPE accelerates premature skin aging using cultured normal human dermal fibroblasts (HDF). We first demonstrated that DPE increases cell senescence marker β-galactosidase activity in HDF. We then found increases in mRNA and protein levels, along with activity of matrix metalloprotease (MMP)-1 and MMP-3, which are associated with skin aging following DPE exposure. We confirmed increases in collagen degradation in HDF treated with DPE. Nicotinamide adenine dinucleotide phosphate (NADPH) oxidase (NOX) is activated by DPE and results in increased ceramide production by sphingomyelinase activation in HDF. We identified that ceramide-1-phosphate (C1P) (produced from ceramide by ceramide kinase activation) activates MMP-1 and MMP-3 through activation of arachidonate cascade, followed by STAT 1- and STAT 3-dependent transcriptional activation.

## 1. Introduction

Both intrinsic and extrinsic factors (i.e., air pollutants and ultraviolet irradiation) drive premature aging [1,2]. Aging is unpreventable in human organs, but premature aging can be slowed down by minimizing the effects of environmental factors on cells/tissues. Indeed, topical agents containing antioxidants have shown some efficacy in attenuating development of visible skin aging symptoms, such as excessive melanogenesis, melanin deposition, and wrinkle formation [3,4].

Diesel particle matter, produced from diesel engines and composed of a various mixture of volatile components, including aldehydes, benzene, polyaromatic hydrocarbons and their derivatives, are major components of air pollutants [2,5]. Polyaromatic hydrocarbons and their derivatives contribute to the development of some diseases, including skin diseases such as contact hypersensitivity and dermatitis, and to the impairment of epidermal keratinocyte function [6,7]. Diesel particle matter also impairs epidermal permeability barrier integrity [8].

In response to stimuli, such as cytokine, angiotensin, and ultraviolet irradiation, nicotinamide adenine dinucleotide phosphate (NADPH) oxidases (NOXs) are activated, which then generate a reactive oxygen species (ROS) [9]. Diesel particulate matter also initiates ROS production through NOX activation [10,11]. ROS generated by NOX plays both a positive (elimination of pathogens) and negative (diseases and aging) role in cells and tissues.

Ceramides (lipids) are an essential component for the epidermal permeability barrier, located in the outermost layer of the epidermis, the stratum corneum, which is comprised of denucleated stratified keratinocyte (corneocytes) and extracellular lipid dominant lamellar structures. In addition, ceramides and their metabolites are lipid mediators that regulate many cellular functions; i.e., proliferation, cell cycle arrest, differentiation, cell death, and cell motility in most mammalian cells, including keratinocyte in the nucleated epidermal layers (stratum basale, stratum spinosum and stratum granulosum) [12,13]. We recently elucidated that diesel particulate extract (DPE)-activated NOX increases cellular ceramide levels by activating sphingomyelinase (SMase), inducing apoptosis in cultured normal human keratinocyte [14].

Epidemiological studies have shown a correlation between levels of pollutants and skin aging symptoms [2]. Decreases in skin elasticity and wrinkle formation are hallmarks of skin aging, which are somehow triggered by structure alterations of the extracellular cellular matrix (ECM) in dermis. Increases in MMP-1 and MMP-3 expression and activity, as well as decreases in production of ECM constituents, contribute to changes in ECM structure and content. Epidermal thickness is reduced by intrinsic epidermal aging. A recent intriguing study demonstrated that a critical role of collagen XVII in maintaining keratinocyte stem cells on basement membranes [15], i.e., decreases in keratinocyte stem cells, influences epidermal thickness and skin regeneration. Production growth factors, including EGF and KGF, as well as responsiveness to growth factors, also determined keratinocyte growth. Yet, the mechanism responsible for thinner epidermis is not completely elucidated. Hence, here we focus on dermal, but not epidermal, aging, and to investigate whether and/or how the lipid mediator, ceramide (and its metabolites) is involved in premature skin aging induced by air pollutants using fibroblasts isolated from normal human dermis. Exposure to DPE increases MMP-1 and MMP-3 production and their activity in HDF. NOX is also activated by DPE, resulting in increased ceramide production by SMase activation in HDF. We have identified that ceramide-1-phosphate (C1P) generated from ceramide by activation of ceramide kinase increases MMP-1 and MMP-3 production, and we further characterized that C1P-mediated activation of arachidonate cascade followed by STAT 1 and STAT 3 activation is a downstream pathway that stimulates MMP-1 and MMP-3 transcription, leading to collagen degradation.

## 2. Results

### 2.1. DPE Increases Cell Senescence

We recently found that DPE induces apoptosis in keratinocyte [14]. In this current study, we investigated the effects of DPE exposure on skin aging. We first optimized concentrations of DPE which do not show cell toxicity. Since up to 20 µg/mL of DPE did not affect cell viability (Figure 1A), concentrations of 0–20 µg/mL of DPE were used in this study. We next examined whether DPE accelerates cell senescence in HDF. β-galactosidase activity, a marker of cell senescence, was increased in HDF following DPE exposure (Figure 1B). These results suggest that DPE accelerates cell senescence in HDF.

### 2.2. DPE Activates MMP-1 and MMP-3

Next we investigated levels of MMP-1 and MMP-3, which are involved in skin aging development by alteration of the extracellular matrix structure [16], in HDF. mRNA expression of MMP-1 and MMP-3 were significantly increased in HDF incubated with DPE in a dose-dependent fashion (Figure 2A,B). We then found dose-dependent significant increases in MMP-1 and MMP-3 protein levels (Figure 2C) and their activities (Figure 2D,E) in cultured medium. These results suggest that DPE could accelerate skin aging by activation of MMP-1 and MMP-3.

### 2.3. DPE Activates NOX

NOX activity was significantly increased in HDF following DPE treatment in a dose-dependent fashion (Figure 3).

### 2.4. Changes in Ceramide and Its Metabolite Profile

We next assessed ceramide and its metabolites (C1P and S1P) levels in HDF treated with DPE. Ceramide levels are significantly increased in HDF following DPE exposure (Figure 4A), in parallel with activation of acidic and neutral SMase (Figure 5A). Ceramide species containing shorter chains of amide-linked fatty acid (C16) amounts were significantly increased in HDF, while longer chain ceramide species were modestly decreased (C24≥) (Figure 4B). Analysis of ceramide metabolites revealed that C1P containing C16 fatty acid levels are significantly increased (Figure 4C,D), while S1P is not changed following DPE treatment (Figure 4E). Ceramide kinase was significantly activated by DPE (Figure 5B). Two isoforms of sphingosine kinase 1 and 2 activity were not changed (Figure 5C,D). These results suggest that DPE increases ceramide production through SMase activation, followed by increased conversion from ceramide to C1P by ceramide kinase activation.

### 2.5. C1P Modulates MMP-1 and MMP-3 Activation

Because both ceramide and C1P are lipid mediators that modulate cellular functions [17,18], we next investigated whether ceramide and/or C1P is (are) responsible for activation of MMP-1 and MMP-3. First, inhibition of acidic and neural SMase by a specific pharmacological inhibitor, imipramine (for acidic SMase) and GW4869 (for neutral SMase), respectively, suppressed DPE-mediated upregulation of both MMP-1 and MMP-3 mRNA expression (Table 1) and protein production (Figure 6). These results suggest that ceramide and/or its metabolite(s) increase(s) MMP-1 and MMP-3 expression.

Next, blocking of C1P synthesis by a specific inhibitor of ceramide kinase, NVP-231 reduced MMP-1 and MMP-3 mRNA expression (Table 1) and protein production (Figure 6). Thus, increases in C1P rather than in ceramide stimulates MMP-1 and MMP-3 production. We further investigated whether activation of MMP-1 and MMP-3 increases collagen degradation. Collagen levels were decreased in HDF exposed to PDE (Figure 7A,B). When SMase and ceramide kinase were inhibited, decreases in collagen were suppressed (Figure 7A,B). These results appear to confirm that DPE-increased C1P simulates MMP-1 and MMP-3, leading to collagen hydrolysis.

### 2.6. Activation of Arachidonate Pathway and STAT1 and STAT3 Are Responsible for C1P Mediated MMP-1 and MMP-3 Activation

Finally, we characterized how C1P stimulates MMP-1 and MMP-3 expression. C1P activates cytosolic phospholipase A2 (cPLA2) that then initiates an arachidonate pathway followed by activation of transcription factors STAT1 and STAT3 [19], leading to an inflammatory response [20]. Moreover, MMP-1 and MMP-3 mRNA expression are increased by STAT1 and STAT 3 activation [21].

Inhibition of cyclooxygenase by acetylsalicylic acid (aspirin) diminished increases in MMP-1 and MMP-3 protein (Figure 6) and decreased in collagen levels in HDF exposed to DPE (Figure 7). STAT1 and STAT3 phosphorylation were also evident in HDF exposed to DPE, while inhibition of sphingomyelinases and ceramide kinase, as well as cyclooxygenase-2, suppressed STAT1 and STAT3 phosphorylation (Figure 8). Thus, C1P stimulates MMP-1 and MMP-3 production through a cPLA2-mediated arachidonate pathway of STAT1 and STAT3 activation.

## 3. Discussion

We demonstrated here that DPE exposure activates MMP-1 and MMP-3, and decreases type 1 collagen levels in HDF. We identified that MMP-1 and MMP-3 activation was initiated by C1P through increases in ceramide production by activation of both acidic and neutral SMase, followed by increased conversion from ceramide to C1P by ceramide kinase activation. We further characterized that C1P-initiated activation of arachidonate pathway followed by STAT1 and STAT3 activation is a downstream mechanism of increased MMP-1 and MMP-3 synthesis (Figure 9).

Prior studies have demonstrated that C1P modulates cellular functions; i.e., adipogenesis, cell migration, and autophagy [17,18]. We also have reported that endoplasmic reticulum (ER) stress-driven increases in C1P promote major epidermal antimicrobial peptide production (human β-defensin [hBD]2 and hBD3) in normal human keratnocyte [22]. Our current study reveals that DPE-increased C1P could accelerate premature skin aging. NOX (which lead to increasing in ceramide and then C1P production) are activated by not only DPE, but also other external and internal stressors, such as ultraviolet irradiation, cigarette smoke, inflammatory cytokines, and epidermal permeability barrier perturbation [9,11,23,24]. Therefore, C1P could play a critical role in skin aging in response to diverse oxidative stressors.

Blocking of sphingomyelinase to reduce C1P production suppresses C1P-mediated MMP-1 and MMP-3 production. Yet, because ceramide is needed to maintain normal epidermal functions, including epidermal permeability barrier formation, specific suppression of C1P rather than ceramide production should be an appropriate therapeutic target to suppress MMP-1 and MMP-3 activation. Inhibition of cPLA2 (which is activated by C1P) to block arachidonate cascade is a potent therapeutic approach to treat inflammatory diseases [25]. However, cPLA2 is activated by multiple pathways and it has not only pathological, but also physiological roles in cells [25,26]. Thus, the inhibition of cPLA2 could lead to unpredictable adverse consequences. Hence, specific modulation of C1P production rather than inhibition of cPLA2, i.e., suppression of C1P production and/or its activity, should be a safer approach to suppressing premature skin aging.

Our prior studies have shown that C16 ceramide is a major backbone constituent of sphingomyelin in skin [27,28]. We found that: (1) C16 ceramide and C16 C1P are significantly increased in HDF following DPE treatment in parallel with SMase activation, and (2) SMase inhibition decreases MMP-1 and MMP-3 activation. Therefore, ceramide derived sphingomyelin, not de novo ceramide synthesis or glycosylceramide (major backbone ceramide containing longer fatty acids) hydrolysis, is the source of ceramide that generates C1P for upregulation of MMP-1 and MMP-3.

Longer chain ceramide species are decreased in HDF following DPE exposure. DPE may affect either/both synthesis and/or activity of ceramide synthase(s) (CerS2 and CerS3, which are synthesized longer chain ceramides) and/or ELOVL1, which synthesizes long chain fatty acids (C24) [29], and results in decreases in longer chain ceramide. Prostaglandin E2 promotes INF-γ production [30]. INF-γ suppresses CerS3 expression [31]. Therefore, DPE could change Cer2, CerS3 and/or ELOVL1 expression in cells, and results in decreases in longer chain ceramide.

S1P is another ceramide metabolite that regulates diverse cell functions, such as cell proliferation, differentiation, apoptosis, angiogenesis and cell motility [32,33,34]. However, S1P levels were decreased in HDF following DPE exposure. Our prior studies showed that ceramide levels were increased, but S1P levels decreased in normal human keratinocyte following DPE treatment [14]. We previously found that both acidic and neutral ceramidase activities are decreased in normal human keratinocyte after UVB irradiation [35]. These results suggest that, under oxidative stress, hydrolysis of ceramide by ceramidase to sphingosine followed by S1P production by sphingosine kinase is likely attenuated. Thus, ceramide could be preferentially metabolized to C1P. It is currently unknown whether and why ceramidase is sensitive to oxidative stress.

Chronic exposure of low levels of DPE may increase in ceramide and its metabolites that are at sub-apoptotic levels, affecting cell proliferation and differentiation. Yet, the mechanism that is responsible for thinner epidermis is still not characterized very well. In addition, the effect of C1P on keratinocyte proliferation and differentiation is unknown. Thus, it is not clear whether C1P contributes to epidermal thinning and regeneration in response to air pollutants and other external stressors. However, we here characterized that C1P activates an arachidonate pathway, leading to stimulation of MMP-1 and MMP-3 mediated collagen hydrolysis. Activation of this arachidonate pathway promotes inflammatory cytokine/chemokine production. Increased secretion of certain inflammatory cytokine/chemokines leads to a senescence-associated secretory phenotype (SASP). C1P may be a driver of SASP.

Penetration of pollutants into skin is not completely elucidated yet either, but a recent study showed that topical diesel engine exhaust penetrated into the dermis and activated MMP-9 in ex vivo cultured human skin [36]. Thus, MMP-1 and MMP-3 could be activated in skin following exposure to DPE.

In conclusion, C1P should be a driver to accelerate premature skin aging through activation of MMP-1 and MMP-3, leading to collagen degradation in dermal fibroblasts following DPE exposure.

## 4. Materials and Methods

### 4.1. Materials

Diesel particulate extract (DPE) used in the present study is Standard Reference Material 1975 (SRM 1975), purchased from the National Institute of Standards and Technology (NIST) (Gaithersburg, MD, USA). Nicotinamide, NADPH, NADP, 2,7-dichlorofluorescin diacetate, GW4869, NVP213 and acetylsalicylic acid were obtained from Sigma-Aldrich (St. Louis, MO, USA). C17-C1P, C17-S1P, S1P (d18:1), sphingosine, ceramides (fatty acid lengths C12, C16, C18, C22, C24, and C24:1), C17-ceramide (d17:1/C18:0), and C12-sphingomyeline sphingosine, ceramides (fatty acid lengths C12, C16, C18, C22, C24, and C24:1) were obtained from Avanti Polar Lipids (Alabaster, AL, USA). Organic solvents for sphingolipid extraction or LC-MS/MS analysis were purchased from Merck (Darmstadt, Germany).

### 4.2. Cell Culture

Normal human dermal fibroblasts (HDF) (Thermo Fisher Scientific, Waltham, MA, USA) were grown as described previously [37,38]. Briefly, HDF were maintained in Dulbecco’s modified Eagle’s medium (DMEM) containing 10% fetal calf serum (FCS) and 1% penicillin/streptomycin. Culture medium was switched to serum-free DMEM one day prior to DPE treatment.

### 4.3. Cell Viability

Cell viability was measured by the water-soluble tetrazolium salt (WST) method using the cell counting kit-8 (CCK-8) assay kit (Dojindo, Kumamoto, Japan), in accordance with the manufacturer’s instructions.

### 4.4. Enzyme Activity

#### 4.4.1. β-Galactosidase

To determine cellular senescence, β-galactosidase activity was measured using senescence β-galactosidase staining kit (Cell signaling, Danvers, MA, USA), in accordance with the manufacturer’s instructions. Images were taken with an inverted microscope (CK-41, Olympus, Tokyo, Japan).

#### 4.4.2. MMP-1 and MMP-3

MMP-1 and MMP-3 activity were assessed by the fluorometric-based method using the MMP-1 or MMP-3 activity assay kit (abcam, Cambridge, MA, USA) according to the manufacturer’s instructions.

#### 4.4.3. NADPH Oxidases

Activity of NADPH oxidase (NOX) was measured by the lucigenin chemiluminescence assay kit using N,N′-Dimethyl-9,9′-bicridium dinitrate (Sigma-Aldrich), in accordance with the manufacturer’s instructions. NADPH oxidases activity was measured by the ratio of NADP+ to NADPH using LC-ESI-MS/MS (API 3200 QTRAP mass, AB/SCIEX, Framingham, MA, USA) by multiple reaction monitoring mode, as we described previously [14].

#### 4.4.4. Sphingomyelinase

Activities of acidic or neutral sphingomyelinase were measured as we described previously [14]. Briefly, cells suspended in assay buffers (acidic sphingomyelinase: 250 mM sodium-acetate, 0.2% Triton X-100, pH 4.5, and neutral sphingomyelinase: 20 mM HEPES, 0.2% Triton X-100, pH 7.4) were incubated with 5 nmol of C12-sphingomyeline for 20 min at 37 °C. The reaction was stopped by the addition of CHCl_3_:CH_3_OH (2:1, *v*/*v*). The organic phases were dried and were resolved in MeOH, and then analyzed by LC-MS/MS. The activities of both sphingomyelinases are expressed as pmol (C12-ceramide) per mg protein per min.

#### 4.4.5. Ceramide Kinase

Ceramide kinase activity was determined as we described previously [39] with modification. Briefly, cell lysates were incubated with 20 μM of C8-ceramide in 20 mM MOPS, pH 7.4, 137 mM NaCl, 5.4 mM KCl, 0.8 mM MgCl_2_, 0.34 mM Na_2_HPO_4_, 5.6 mM glucose, 0.44 mM KH_2_PO_4_, 4.2 mM NaHCO_3_, 1.9 mM CaCl_2_, and 0.1% BSA with 1 mM ATP at 37 °C for 30 min. Enzyme reactions were terminated by the addition of cold MeOH containing 30 pmoles of d17:1/C18:0 ceramide as the internal standard and the activity was quantified by LC-MS/MS. The activity of ceramide kinase is expressed as pmol (C8-ceramide) per mg protein per min.

#### 4.4.6. Sphingosine Kinase

Sphingosine kinase 1 and Sphingosine kinase 2 activities were assessed, as we described previously [14,40]. Briefly, cell lysates were incubated with 200 μM C17-Sphingosine as a substrate. To assay each isoform of sphingosine kinase activity, 0.5% Triton X-100 or 1 M KCl for sphingosine kinase 1 and sphingosine kinase 2, respectively, were added into assay buffer and then incubated at 37 °C for 30 min. Enzyme reactions were terminated by the addition of CHCl_3_:MeOH:HCl (8:4:3, *v*/*v*/*v*). C17-dihydrosphingosine-1-phosphate (100 pmol) was added as an internal standard. The organic phage was separated by addition of CHCl_3_, dried and redissolved in MeOH, and then analyzed by LC-MS/MS. The activity of Sphingosine kinase was expressed as C17-S1P pmol per mg protein per min.

### 4.5. Detection of Cellular Reactive Oxygen Species

Production of cellular reactive oxygen species (ROS), including superoxide (O_2_^−^) and hydrogen peroxide (H_2_O_2_) was detected by the oxidant-sensing probe 2,7-dichlorodihydrofluorescein diacetate (DCFH-DA) (abcam), as described previously [38]. ROS production was analyzed using a fluorescence microscopy (Eclipse Ti-U; Nikon Corporation, Tokyo, Japan) and fluorospectrophotometer (Molecular devices M2e, Molecular Devices, Sunnyvale, CA, USA) with 485 nm of excitation and 520 nm of emission filters and was expressed as a fluorescence intensity (a.u.).

### 4.6. Measurement of Ceramide, Ceramide-1-Phosphate and Sphingosine-1-Phosphate

Total lipids were extracted from cells by the method of Bligh and Dyer (1959), with modification, as described previously [40,41]. Ceramide, Ceramide-1-phosphate (C1P) and Sphingosine-1-Phosphate (S1P) were quantitated using LC-ESI-MS/MS (API 3200 QTRAP mass), as described previously [40,41,42]. The ceramide MS/MS transitions (*m*/*z*) were 510→264 for C14-ceramide, 538→264 for C16-ceramide, 552→264 for C17-ceramide, 566→264 for C18-ceramide, 594→264 for C20-ceramide, 648→264 for C24:1-ceramide, 650→264 for C24-ceramide, 676→264 for C26:1-ceramide, and 678→264 for C26-ceramide, respectively. The C1P MS/MS transitions (*m*/*z*) were 590→264 for C14-C1P, 618→264 for C16- C14-C1P, 646→264 for C18- C14-C1P, 674→264 for C20- C14-C1P, 728→264 for C24:1- C14-C1P, 730→264 for C24- C14-C1P, 756→264 for C26:1- C14-C1P, and 758→264 for C26- C14-C1P, respectively. The S1P MS/MS transitions (*m*/*z*) were 366→250 for C17 S1P as an internal standard and 380→264 for C18 sphingosine 1-phosphate, respectively. Data were acquired using Analyst 1.7.1 software (Applied Biosystems, Foster City, CA, USA). Sphingolipid levels are expressed as pmol per mg protein.

### 4.7. Western Blot Analysis

Protein levels of MMP-1 and MMP-3 were assessed by Western blot analysis, as we described previously [40]. Briefly, cell lysates, prepared in radioimmunoprecipitation assay buffer, were resolved by electrophoresis on 4–12% Bis-Tris protein gel (Invitrogen, Carlsbad, CA, USA). Resultant bands were blotted onto polyvinylidene difluoride membranes, probed with anti-MMP-1 and MMP-3 (Santa Cruz Biotechnology, Dallas, TX, USA), anti-phospho-STAT1 (Tyr-701) anti-phospho-STAT3 (Tyr-705) (Cell Signaling Technology, Boston, MA, USA), and anti-human β-actin (Sigma-Aldrich), and detected using enhanced chemiluminescence (Thermo Fisher Scientific, Waltham, MA, USA). The intensity of bands was measured with a LAS-3000 (Fujifilm, Tokyo, Japan).

### 4.8. Quantification RT-PCR Analysis

Relative mRNA expression was assessed by quantitative RT-PCR (*q*RT-PCR), as we described previously [40]. Total RNA was isolated from cell lysates using RNeasy mini kit (Qiagen, Germantown, MD), followed by preparation of cDNA using SensiFAST^TM^ cDNA synthesis kit (Bioline/Meridian, Taunton, MA, USA). The following primer sets were used: MMP-1, 5′-ATGAAGCAGCCCAGATGTGGAG-3′ and 5′-TGGTCCACATCTGCTCTTGGCA-3′; MMP-3, 5′-CACTCACAGACCTGACTCGGTT-3′ and 5′- AAGCAGGATCACAGTTGGCTGG-3′; GAPDH, 5′-GGAGTCAACGGATTTGGTCGTA-3′ and 5′-GCAACAATATCCACTTTACCAGAGTTAA-3′. The thermal cycling conditions were 95 °C for 10 min, 95 °C for 15 s, 60 °C for 15 s, and 72 °C for 15 s, repeated 40 times on ABI Prism 7500 (Applied Biosystems, Waltham, MA, USA). mRNA expression was normalized to levels of GAPDH. Values shown represent mean (±SD) for three independent assays.

### 4.9. Collagen Type I Analysis

Collagen type I content was measured by total collagen type I assay kit (Biovision, Milpitas, CA, USA) according to the manufacturer’s instructions. Collagen type I was probed with anti-collagen type I (abcam).

### 4.10. Statistical Analyses

Results were expressed as the mean ± standard deviation (SD). Statistical analyses were performed using the Prism Version 6.0 Software (GraphPad Software, San Diego, CA, USA). Significance between groups was determined by unpaired Student’s *t*-test. The *p* values were set at <0.01.

## Figures and Tables

**Figure 1 ijms-23-02691-f001:**
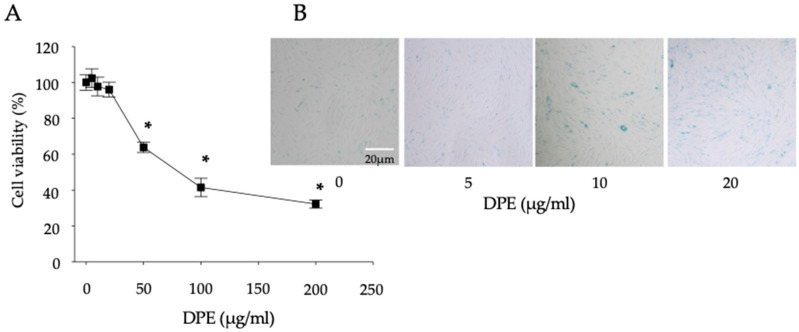
Cell viability and β-galactosidase activity. HDF were incubated with DPE for 24 h. Cell viability was measured by water-soluble tetrazolium salt cell quantification method (**A**). Cells positive for β-galactosidase activity were stained with blue-green (**B**). All values are mean ± SD (*n* = 3). Statistical significance was calculated using the unpaired Student’s *t*-test, and significance was defined as * *p* < 0.01 vs. vehicle control. See details in Section 4.

**Figure 2 ijms-23-02691-f002:**
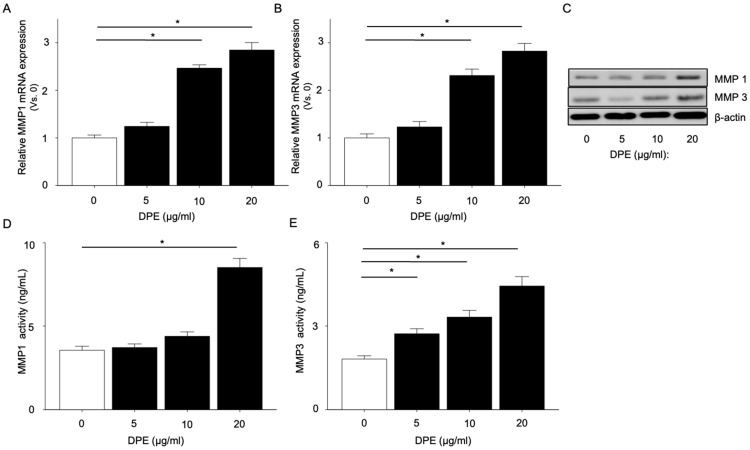
MMP-1 and MMP-3 mRNA, protein levels, and activity. HDF were incubated with DPE or vehicle for 24 h. MMP-1 and MMP-3 mRNA (**A**,**B**), protein expression (**C**), and enzymatic activity (**D**,**E**) were measured by *q*RT-PCR, Western blot, or the fluorometric-based assay, respectively. All values are mean ± SD (*n* = 3). * *p* < 0.01 vs. vehicle control. See details in Section 4.

**Figure 3 ijms-23-02691-f003:**
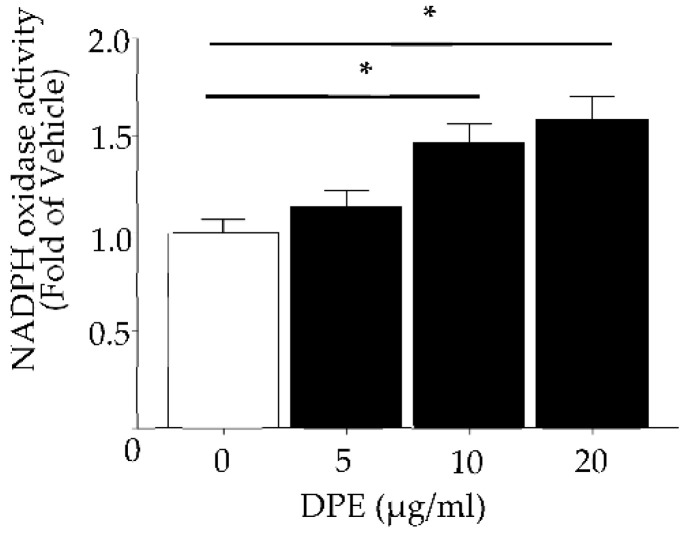
NADPH Oxidase activity. HDF were incubated with DPE or vehicle for 24 h. Activity of NAPDH oxidases (NOX) was determined by measuring the ratio of NADP^+^ to NADPH with LC-ESI-MS/MS (API 3200 QTRAP mass, AB/SCIEX) by multiple-reaction monitoring mode (MRM). All values are mean ± SD (*n* = 3). * *p* < 0.01 vs. vehicle control. See details in Section 4.

**Figure 4 ijms-23-02691-f004:**
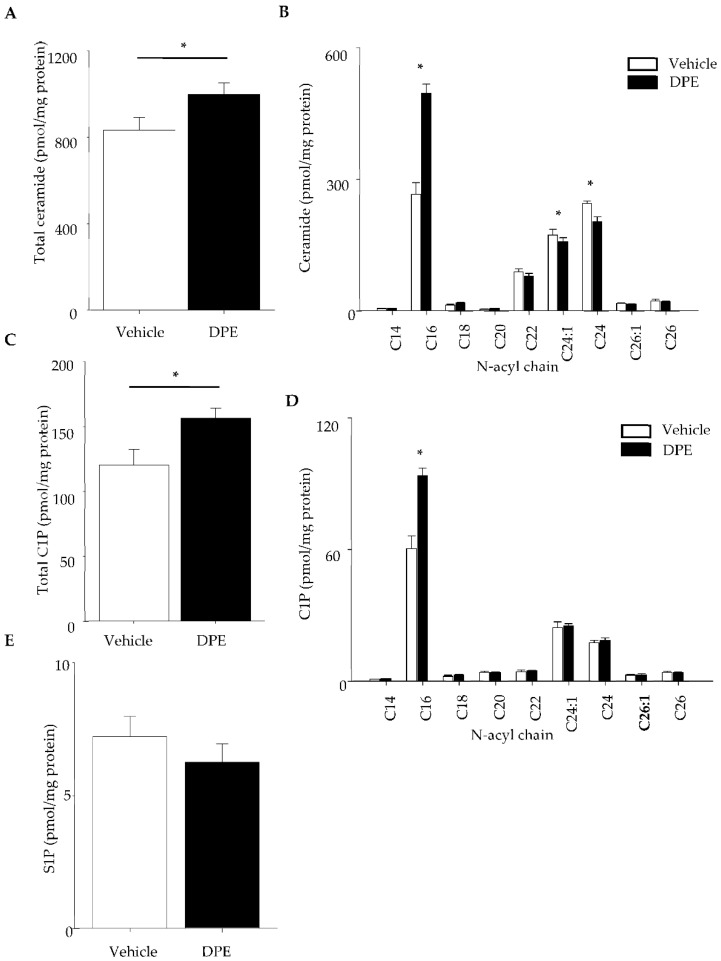
Total ceramide (**A**), C1P (**C**), and S1P (**E**) content. Ceramide (**B**) and C1P (**D**) species of different amide-linked fatty acid content. HDF were treated with DPE for 24 h, followed by ex-traction of lipids. Ceramide, C1P and S1P content were measured by LC-ESI-MS/MS. All values are mean ± SD (*n* = 3). * *p* < vs. vehicle control. See details in Section 4.

**Figure 5 ijms-23-02691-f005:**
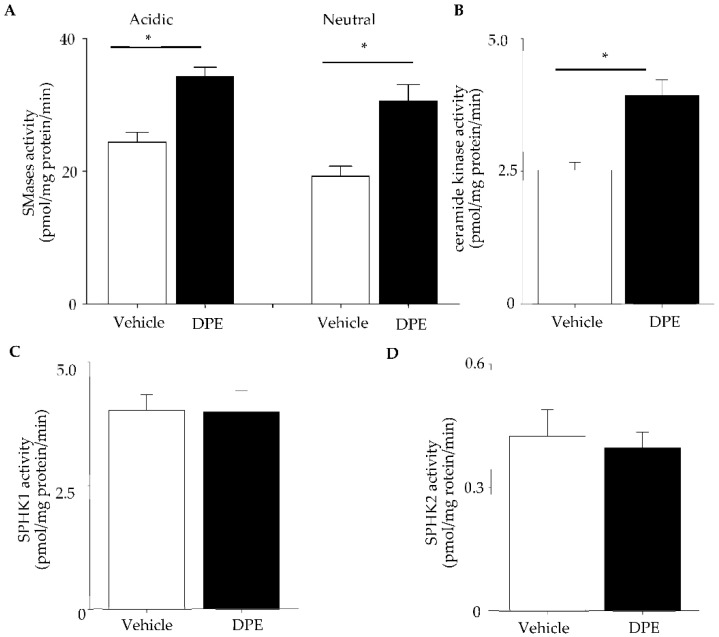
Sphingomyelinases, ceramide kinase, and sphingosine kinase activity. HDF were incubated with 20 µg/mL of DPE or vehicle for 24 h. Acidic and neutral SMase (**A**), ceramide kinase (**B**), sphingosine kinase 1 (**C**), and sphingosine kinase 2 (**D**) were assessed by LC-ESI-MS/MS. All values are mean ± SD (*n* = 3). * *p* < 0.01 vs. vehicle control. See details in Section 4.

**Figure 6 ijms-23-02691-f006:**
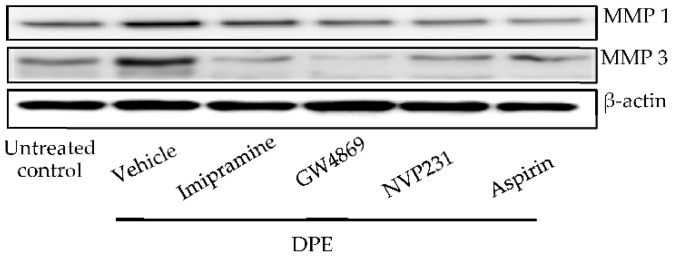
MMP-1 and MMP-3 protein expression. HDF were incubated with or without imipramine (acidic SMase inhibitor), GW4869 (neutral SMase inhibitor), VP231 (ceramide inhibitor), or Aspirin (cyclooxygenase-2 [COX-2] inhibitor) for 30 min, and then incubated with 20 µg/mL of DPE for 24 h. MMP-1 and MMP-3 protein expressions were assessed by Western blot assay.

**Figure 7 ijms-23-02691-f007:**
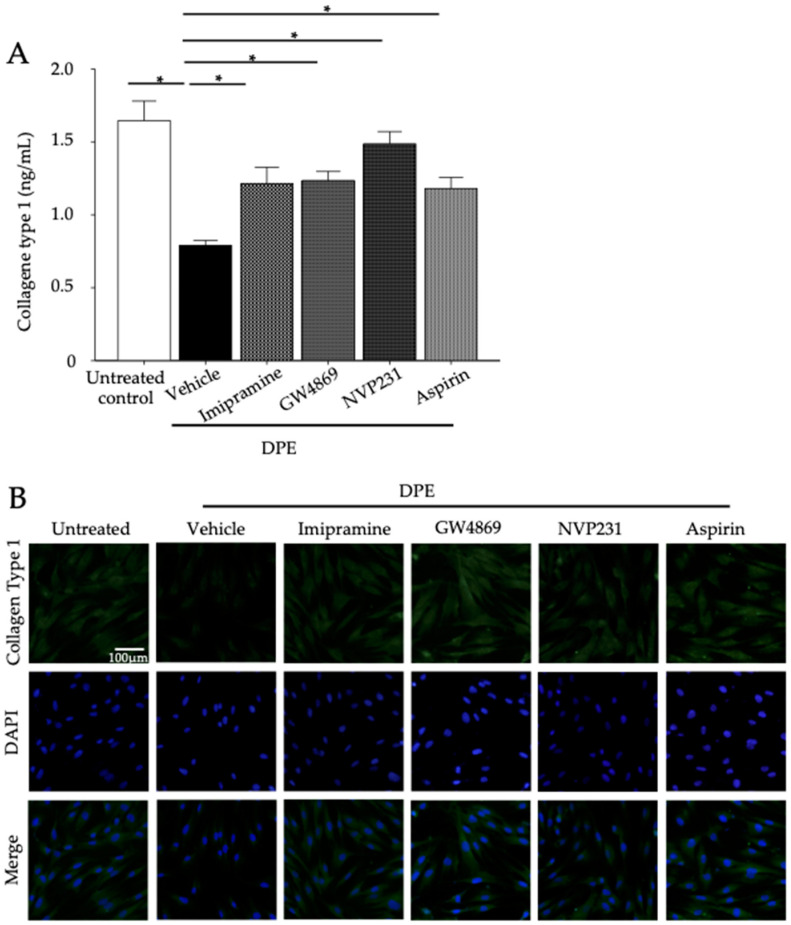
Collagen type I content. HDF were incubated with or without imipramine (acidic SMase inhibitor), GW4869 (neutral SMase inhibitor), VP231 (ceramide kinase inhibitor), or Aspirin (cyclooxygenase-2 [COX-2] inhibitor) for 30 min, and then incubated with 20 µg/mL of DPE for 24 h. Collagen type I content was measured using ELISA (**A**). All values are mean ± SD (*n* = 3). * *p* < 0.01 vs. vehicle control. Immunofluorescence staining of collagen type I (**B**). See details in Section 4.

**Figure 8 ijms-23-02691-f008:**
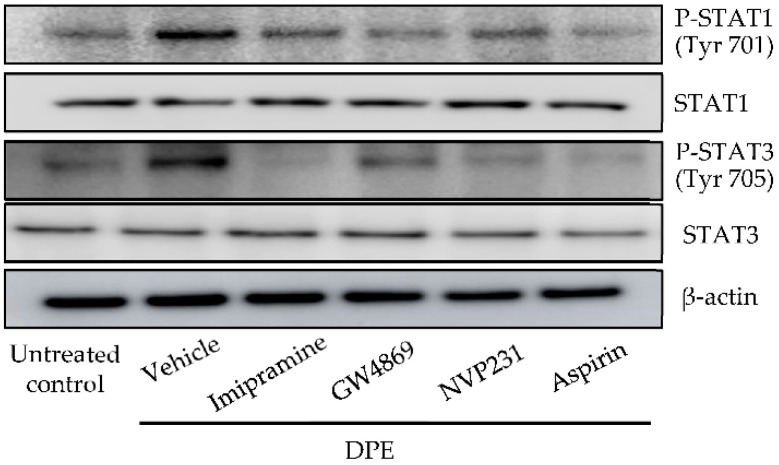
Phosphorylation of STAT1 and STAT3. HDF were incubated with or without imipramine (acidic sphingomyelin inhibitor), GW4869 (neutral sphingomyelin inhibitor) and VP231 (ceramide kinase inhibitor) for 30 min, and then incubated with 20 µg/mL of DPE for 24 h. Both total and phosphorylated levels of STAT1 (Tyr-701) or STAT3 (Tyr-705) were assessed by Western blot assay. See details in Section 4.

**Figure 9 ijms-23-02691-f009:**
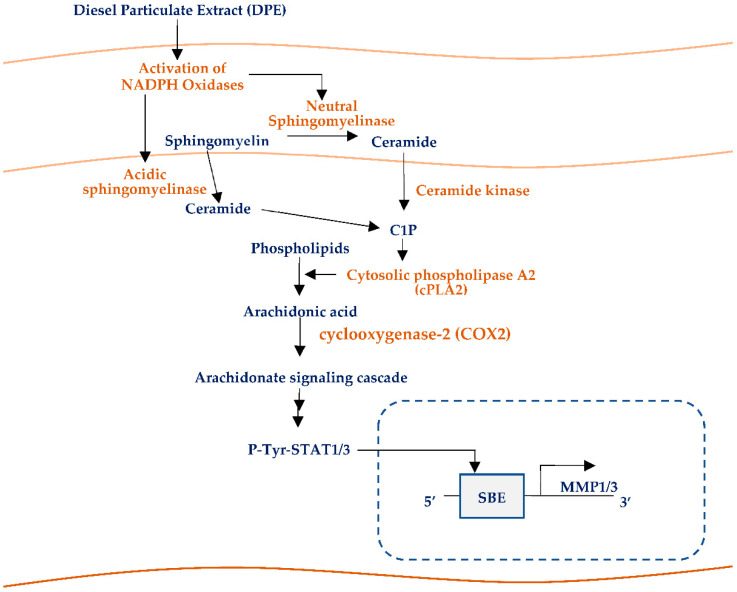
Acceleration of premature skin aging by DPE via NADPH) oxidases-mediated C1P pathway.

**Table 1 ijms-23-02691-t001:** MMP-1 and MMP-3 mRNA expression.

Treatment	Relative mRNA Expression vs. Vehicle Control	
	MMP1	MMP3
Control	1.00 ± 0.12	1.00 ± 0.05
Imipramine	1.04 ± 0.09	0.98 ± 0.08
GW4869	1.11 ± 0.14	0.98 ± 0.06
NVP231	1.16 ± 0.19	1.01 ± 0.13
DPE	3.34 ± 0.34	3.20 ± 0.49
DPE + Imipramine	1.86 ± 0.21 *	1.55 ± 0.15 *
DPE + GW4869	1.66 ± 0.17 *	1.44 ± 0.09 *
DPE + NVP231	1.63 ± 0.18 *	1.50 ± 0.09 *

* *p* < 0.01 vs. DPE.

## Data Availability

Data are not available.

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
