# Peer review of "Diesel Particulate Extract Accelerates Premature Skin Aging in Human Fibroblasts via Ceramide-1-Phosphate-Mediated Signaling Pathway"

_ijms, 2022, doi:10.3390/ijms23052691_

Round 1

Reviewer 1 Report

This study examined the effects of diesel particulate extract (DPE) on premature aging in cultured human dermal fibroblasts. DPE increased beta-galactosidase activity, indicating accelerated cell senescence. DPE also increased the expression and activities of MMP-1 and MMP-3, which are involved in skin aging. A series of studies using several inhibitors have shown that the mechanism requires ceramide-1-phosphate (C1P) produced by activating NADPH oxidases (NOX). It has also been shown that C1P increases MMP-1 and MMP-3 production through activation of arachidonate cascade, followed by STAT1 and STAT3-dependent transcriptional activation. How skin aging occurs and is prevented has attracted much attention in recent years. This study, although partly, clarifies the mechanism of skin aging and thus is fascinating. However, the reviewer has several questions and comments below.

1. The authors have previously shown that DPE increases ceramides by NOX activation, inducing apoptosis in cultured normal human keratinocytes. The reviewers did not know why the authors focused on fibroblasts in the present study. Please explain the rationale in the introduction.

2. Regarding the effect of drugs on DPE-induced expression of MMPs: The protein expression levels appear to be variable between groups and not yet quantified. In addition, it may be worth considering the effects on mRNA levels.

3. The role of ceramides and their metabolites in skin aging should be further discussed. For example, what is the role of keratinocyte C1P in skin aging? What role do keratinocytes and fibroblasts play in skin aging when treated with DPE or other stressors in vivo?

4. Ceramides are essential for the skin barrier, so it is not reasonable to inhibit ceramide production. Please discuss what molecules should be targeted in order to prevent skin aging.

5. This study used up to 20 µg/mL DPE because it did not exhibit cell toxicity. Does the concentration reflect the actual exposure concentration in the skin?

6. DPE increased the amounts of ceramide species containing shorter chains (C16) of amide-linked fatty acid while decreasing longer chain ceramides (C24:1 and C24:0). Please explain why the decrease in long-chain ceramides occurs and whether it affects cellular responses.

7. Page 5, lane 118-119 and Page 10, and 212-213: Although the authors describe that S1P was decreased by DPE treatment, it seems insignificant. Please revise it.

8. The fluorescence intensity of DAPI appears different between the groups.

9. The descriptions of many methods are lacking. Please describe how the lipids were extracted.

Author Response

We thank you for reviewing our. We have revised it, incorporating reviewers’ relevant comments and suggestions, and including additional data.

Reviewer 1

Comment 1: The authors have previously shown that DPE increases ceramides by NOX activation, inducing apoptosis in cultured normal human keratinocytes. The reviewers did not know why the authors focused on fibroblasts in the present study. Please explain the rationale in the introduction.

Response: Decreases in skin elasticity and wrinkle formation are hallmarks of skin aging and are somehow triggered by structure alterations of the extracellular cellular matrix (ECM) in dermis. Increases in MMP-1 and MMP-3 expression and activity, as well as decreases in production of ECM constituents, contribute to changes in ECM structure and content. Epidermal thickness is reduced by intrinsic epidermal aging. A recent intriguing study demonstrated that a critical role of collagen XVII in maintaining keratinocyte stem cells on basement membranes (Nature, 2019, 568: 344-50), i.e., decreases in keratinocyte stem cells, influences epidermal thickness and skin regeneration. Production growth factors, including EGF and KGF, as well as responsiveness to growth factors, also determined keratinocyte growth. Yet, the mechanism responsible for thinner epidermis is not completely elucidated. Hence, in this study, we focus on dermal, not epidermal aging, and we investigated whether and/or how the lipid mediator, ceramide (and its metabolites) is involved in premature dermal aging induced by air pollutants using fibroblasts isolated from normal human dermis.

We include these statements in our revised Introduction section.

We discovered that the ceramide metabolite, C1P is involved in dermal skin aging. Thus, we now can explore involvement of ceramide and its metabolites in epidermal and subcutaneous tissue (hair follicle, sebaceous gland) aging. 

Comment 2: Regarding the effect of drugs on DPE-induced expression of MMPs: The protein expression levels appear to be variable between groups and not yet quantified. In addition, it may be worth considering the effects on mRNA levels.

Response: We considered conducting semi-quantification of western blot data, but it is not very accurate, so we reported only western blot photography. As suggested by this reviewer, we have now included mRNA levels assessed by qRT-PCR in our revision (Table 1). Levels of mRNA expression of MMP-1 and MMP-3 are consistent with results of western blot testing. We further assured C1P modulates MMP-1 and MMP-3 expression in cells in response to DPE.

Comment 3: The role of ceramides and their metabolites in skin aging should be further discussed. For example, what is the role of keratinocyte C1P in skin aging? What role do keratinocytes and fibroblasts play in skin aging when treated with DPE or other stressors in vivo?

Response: Chronic exposure of low levels of DPE may increase in ceramide and its metabolites that are at sub-apoptotic levels, affecting cell proliferation and differentiation. Yet, the mechanism that is responsible for thinner epidermis is still not characterized very well. In addition, the effect of C1P on keratinocyte proliferation and differentiation is unknown. Thus, it is not clear whether C1P contributes to epidermal thinning and regeneration in response to air pollutants and other external stressors. However, we here characterized that C1P activates an arachidonate pathway, leading to stimulation of MMP-1 and MMP-3 mediated collagen hydrolysis. Activation of this arachidonate pathway promotes inflammatory cytokine/chemokine production. Increased secretion of certain inflammatory cytokine/chemokines leads to a senescence-associated secretory phenotype (SASP). C1P may be a driver of SASP.

Comment 4: Ceramides are essential for the skin barrier, so it is not reasonable to inhibit ceramide production. Please discuss what molecules should be targeted in order to prevent skin aging.

Response: Inhibition of ceramide production by blocking sphingomyelinase activity is a strategy to decrease C1P production. Yet, because ceramide is needed to maintain normal epidermal functions, including epidermal permeability barrier formation, specific suppression of C1P rather than ceramide production should be an appropriate therapeutic target to suppress MMP-1 and MMP-3 activation.

Comment 5:  This study used up to 20 µg/mL DPE because it did not exhibit cell toxicity. Does the concentration reflect the actual exposure concentration in the skin?

As described in the Discussion section (paragraph 6), prior studies demonstrated that MMP-9 levels are increased in skin exposed to air pollutants (Novel Spray Dried Algae-Rosemary Particles Attenuate Pollution-Induced Skin DamageMecules 2021 26:3781. doi: 10.3390/molecules26133781). In this study, 12mm diameter skin is exposed to air pollutants generated by diesel engine Kubota RTV-X900 diesel engine (3-cylinder, 4-cycle diesel with overhead valves, 1123 cc that has 24.8 HP at 3000 rpm). Estimated total amount of PM (particulate matter) is over 100 mg.

Response: The penetration rate of chemicals from PM is not reported in this study. Others reported that absorption of topical 2,3,7,8-tetrachlorodibenzo-p-dioxin (TCDD) into rat dermis is 0.4% (Banks YB, Birnbaum LS. Toxicol Appl Pharmacol, 107: 302-10, 1991). Note, TCDD is a component of DPE. Other components such as aromatic hydrocarbons are thought to be equivalent and more hydrophobic (J Environ Monit 2005, 7: 983-88). The molecular weight of all components is below 500 and hydrophobic, suggesting that DPE components can penetrate skin.

If 0.1% of chemicals in DPE penetrated the dermis, 100µg/1.16 square cm (12mm diameter) = 100µg/cm2. Hence, 10-20 µg/ml of DPE is a feasible concentration for in vivo skin imitating skin exposure to air pollutants. Indeed, epidemiological studies (cited in this manuscript) already demonstrated that skin aging levels are associated with air pollutant penetration.

  1. DPE increased the amounts of ceramide species containing shorter chains (C16) of amide-linked fatty acid while decreasing longer chain ceramides (C24:1 and C24:0). Please explain why the decrease in long-chain ceramides occurs and whether it affects cellular responses.

Response: As described in our manuscript, C16 ceramide provides the predominant backbone of sphingomyelin structure. Thus, increases in sphingomyelin hydrolysis could increase C16 ceramide. DPE may affect either/both synthesis and/or activity of ceramide synthase(s) (CerS2 and CerS3, which are synthesized long-chain ceramides) and/or ELOVL1, which synthesizes long chain fatty acids (C24), and results in decreases in longer chain ceramide by C1P-mediated activation of arachidonate cascade. Prostaglandin E2 promotes INF-gamma production (J Periodontol, 2003, 74: 771-9), and INF-gamma suppresses CerS3 expression (J Invest Dermatol 2014;134: 712-8).

Therefore, DPE could change Cer2, CerS3 and/or ELOVL1 expression in cells, resulting in decreased longer-chain ceramide.

Decreases in longer chain ceramide may affect glycosyl sphingolipid profiles, galactosylceramide (Note, the major monoglycosylceramide in fibroblast is galactocylceramide) and polyglycosylceramides (asialo- and sialoglcosphingolipids). These glycosphingolipids are plasma membrane constituents, and some polyglycosylsphingolipids are lipid mediators that alter certain cellular functions. Thus, decreases in longer chain ceramide may affect cellular functions.

We have included why long-chain ceramides are decreased in cells following DPE exposure.

  1. Page 5, lane 118-119 and Page 10, and 212-213: Although the authors describe that S1P was decreased by DPE treatment, it seems insignificant. Please revise it.

Response: Following reviewer’s comment, we revised this sentence.

“S1P levels are not significantly changed….”

  1. The fluorescence intensity of DAPI appears different between the groups.

Response: As indicated by this reviewer, DAPI intensity is relatively weak in control, GW486 and NVP231. If the numbers of DAPI positive cells are different, cell growth or cell death are indicated, but total numbers were not different among these groups. The purpose of this immunostaining is to assure ELISA data in evaluating collagen type 1 levels. Hence, fluorescence intensity of DAPI does not affect our conclusions. Nevertheless, following this reviewer’s suggestion, we replaced some DAPI images.

  1. The descriptions of many methods are lacking. Please describe how the lipids were extracted.

Response: Following this reviewer’s comment, we have provided more detailed experimental methods for lipid extraction and our enzyme assay method in our revision.

Reviewer 2 Report

The authors conducted an impressive investigation and established that exposure to Diesel Particulate Extract (DPE) accelerates the aging process of the skin. Prior studies have linked diesel particulate to an inflammatory response, and the relationship between inflammation and aging is well-established. In this manuscript, the authors demonstrate an increase in the key component ceramide-1-phosphate (C1P) as a result of DPE exposure and described the pathway for CIP-mediated collagen degradation through activation of MMP-1 and MMP-3 via the arachidonate cascade.

In my opinion, this is a well-written article, and the authors presented their research findings clearly and concisely. The findings can be significant for progress in therapeutics approaches to tackle premature aging via modulating C1P activity.

I would recommend that the authors briefly outline potential research directions for C1P inhibition in the treatment of premature aging.

  1. If there are any possible drawbacks to suppressing C1P due to involvement in other critical regulatory functions?
  2. The authors indicate that inhibiting C1P with cPLA2 may have detrimental repercussions. Then, what further strategies would the authors recommend to inhibit C1P for combating premature aging?

Author Response

We thank you for reviewing our. We have revised it, incorporating reviewers’ relevant comments and suggestions, and including additional data.

Comment 1: I would recommend that the authors briefly outline potential research directions for C1P inhibition in the treatment of premature aging.

If there are any possible drawbacks to suppressing C1P due to involvement in other critical regulatory functions?

Response: C1P has diverse biological roles in cells as described in our manuscript. In skin, cell motility is essential for wound healing. Hence, delayed wound healing is a potential risk by suppressing C1P production in skin. For clinical application of C1P, including for skin care purposes, an in vivo study to investigate optimal levels of suppression of C1P is essential. We have included these statements in our Discussion.

Comment 2: The authors indicate that inhibiting C1P with cPLA2 may have detrimental repercussions. Then, what further strategies would the authors recommend to inhibit C1P for combating premature aging?

Response: As we already wrote in our Discussion, “Thus, the inhibition of cPLA2 could lead to unpredictable adverse consequences. Hence, specific modulation of C1P production rather than inhibition of cPLA2; i.e., suppression of C1P production and/or its activity, should be a safer approach to suppressing premature skin aging.”